# Role of Lanthanides and Bilayer Fe_2_As_2_ in the Electronic Properties of Rb*Ln*_2_Fe_4_As_4_O_2_ (*Ln* = Gd, Tb, and Dy) Superconductors

**DOI:** 10.3390/ma16114123

**Published:** 2023-06-01

**Authors:** Yi-Na Huang, Da-Yong Liu, Hong-Ying Mei, Li Han, Huan-Ping Yang

**Affiliations:** 1Department of Physics, School of Science, Zhejiang University of Science and Technology, Hangzhou 310023, China; 2Department of Physics, School of Sciences, Nantong University, Nantong 226019, China; 3Henan Provincial Key Laboratory of Smart Lighting, School of Information Engineering, Huanghuai University, Zhumadian 463000, China; 4Qilu Institute of Technology, School of Chemical and Biological Engineering, Jinan 250200, China

**Keywords:** iron-based superconductor, electronic structure, magnetism

## Abstract

The superconducting transition temperatures (T${}_{c}$) of RbGd${}_{2}$Fe${}_{4}$As${}_{4}$O${}_{2}$, RbTb${}_{2}$Fe${}_{4}$As${}_{4}$O${}_{2}$, and RbDy${}_{2}$Fe${}_{4}$As${}_{4}$O${}_{2}$ are 35 K, 34.7 K, and 34.3 K without doping, respectively. For the first time, we have studied the high-temperature nonmagnetic state and the low-temperature magnetic ground state of 12442 materials, RbTb${}_{2}$Fe${}_{4}$As${}_{4}$O${}_{2}$ and RbDy${}_{2}$Fe${}_{4}$As${}_{4}$O${}_{2}$, using first principles calculations and comparing them with RbGd${}_{2}$Fe${}_{4}$As${}_{4}$O${}_{2}$. We also performed a detailed study of the effects of lanthanides and bilayer Fe${}_{2}$As${}_{2}$. We predict that the ground state of Rb*Ln*${}_{2}$Fe${}_{4}$As${}_{4}$O${}_{2}$ (*Ln* = Gd, Tb, and Dy) is spin-density-wave-type, in-plane, striped antiferromagnets, and the magnetic moment around each Fe atom is about 2 $\mu_{B}$. We also found that the structural differences caused by the simple ionic radius have little effect on the properties of these three materials. Different lanthanide elements themselves play a major role in the electronic properties of the materials. It can be confirmed that the effect of Gd on Rb*Ln*${}_{2}$Fe${}_{4}$As${}_{4}$O${}_{2}$ is indeed different from that of Tb and Dy, and the presence of Gd is more conducive to interlayer electron transfer. This means that Gd can transfer more electrons from the GdO layer to the FeAs layer compared to Tb and Dy. Therefore, RbGd${}_{2}$Fe${}_{4}$As${}_{4}$O${}_{2}$ has a stronger internal coupling strength of the bilayer Fe${}_{2}$As${}_{2}$ layer. This can explain why the T${}_{c}$ of RbGd${}_{2}$Fe${}_{4}$As${}_{4}$O${}_{2}$ is slightly higher than that of RbTb${}_{2}$Fe${}_{4}$As${}_{4}$O${}_{2}$ and RbDy${}_{2}$Fe${}_{4}$As${}_{4}$O${}_{2}$.

## 1. Introduction

Finding new superconductors is an important and challenging task. After long-term efforts, especially in the past three decades, two basic routes for finding new superconducting materials have been discovered. The first is the chemical doping of potentially superconducting parent compounds, and the other is to synthesize new compounds containing two-dimensional superconducting active layers through structural design. Well-known examples are copper-based and iron-based high-temperature superconductors whose conductive active layers are CuO${}_{2}$ planes and Fe${}_{2}$*X*${}_{2}$ layers, respectively. Impressively, these far-reaching discoveries were made by chemical doping, and the structures of these high-temperature superconductors can be explained well by their crystal chemistry. Optimized doping levels and building block layers may set new records related to the superconducting transition temperature T${}_{c}$ [1]. However, it is worth noting that superconductivity can also occur in undoped materials, where charge carriers are induced by intrinsic charge transfer through self-doping.

In recent years, the discovery of iron-based superconductors (FeSCs) has sparked a great deal of research activity. So far, dozens of FeSC crystal types have been discovered. Representative systems (with numerical abbreviations corresponding to their chemical compositions) that have been investigate in depth include (i) FeSe (11) [2], (ii) LiFeAs (111) [3], (iii) BaFe${}_{2}$As${}_{2}$ (122) [4], (iv) K${}_{x}$Fe${}_{2-y}$Se${}_{2}$ (122*) [5], and (v) LaFeAsO (1111) [6,7]. All of these compounds consist of fluorite-like layers of Fe${}_{2}$*X*${}_{2}$ (*X* = phosphorus group or chalcogenide); thus, the Fe${}_{2}$*X*${}_{2}$ layer is considered a key building block for Fe-based superconductivity. This is further supported by the fact that some other compounds containing the same Fe${}_{2}$*X*${}_{2}$ layer (but with a different barrier layer in between) also show superconductivity or become superconductors after proper chemical doping, with a relatively high T${}_{c}$. The 1111, 122, 111, and 11 iron-based high-temperature superconductors that have been discovered so far are all quasi-two-dimensional layered structures with Fe${}_{2}$*X*${}_{2}$ layers. It was found that by adding calcium-state mineral-type layers between Fe${}_{2}$As${}_{2}$ layers, the so-called 32522 and 42622 new systems can be obtained, and their typical representatives are Sr${}_{3}$Sc${}_{2}$O${}_{5}$Fe${}_{2}$As${}_{2}$ [8] and Sr${}_{4}$V${}_{2}$O${}_{6}$Fe${}_{2}$As${}_{2}$ [9]. It is also possible to insert a less symmetrical spacer layer between the Fe${}_{2}$As${}_{2}$ layers.

So far, Cao Guanghan’s group [10] has designed several new structures containing Fe${}_{2}$X${}_{2}$ layers. One of the candidate structures, “KLaFe${}_{4}$*X*${}_{4}$”, was realized in the form of *AeA*Fe${}_{4}$As${}_{4}$ (*Ae* = K, Rb, and Cs; *A* = Ca and Sr) [11] and *Ak*EuFe${}_{4}$As${}_{4}$ (*Ak*=Rb and Cs) [12]. Another proposed structure, *Ln*${}_{3}$Fe${}_{4}$X${}_{4}$Z${}_{2}$ [13], was realized in KCa${}_{2}$Fe${}_{4}$As${}_{4}$F${}_{2}$ (hereafter referred to as 12442). At the same time, they also synthesized *Ln*Ca${}_{2}$Fe${}_{4}$As${}_{4}$F${}_{2}$ (where *Ln* = Rb and Cs) [14], which are two sister compounds of KCa${}_{2}$Fe${}_{4}$As${}_{4}$F${}_{2}$, and they can be regarded as the interbreeding body of *Ln*Fe${}_{2}$As${}_{2}$ (*Ln* = Rb and Cs) and CaFeAsF. This crystal structure has bilayer Fe${}_{2}$As${}_{2}$ layers separated by insulating fluorite-type Ca${}_{2}$F${}_{2}$ plates, similar to the case of double CuO${}_{2}$ planes in cuprate superconductors. Its crystallization parameters were measured. Physical property measurements show that the bulk RbGd${}_{2}$Fe${}_{4}$As${}_{4}$O${}_{2}$, RbTb${}_{2}$Fe${}_{4}$As${}_{4}$O${}_{2}$, and RbDy${}_{2}$Fe${}_{4}$As${}_{4}$O${}_{2}$ have T${}_{c}$ values of 35 K, 34.7 K, and 34.3 K, respectively [15]. The development of novel superconductors whose bulk superconducting structures match the Fe${}_{2}$*X*${}_{2}$ layers and satisfy thermodynamic or at least kinetic stability is important and challenging.

The 12442 material is synthesized from 1111 and 122 materials, *AkLn*${}_{2}$Fe${}_{4}$As${}_{4}$O${}_{2}$ (*Ak* = K and Cs; *Ln* = Lanthanides) [15] iron-based superconducting material. This material exhibits superconductivity without extrinsic doping and without spin density wave (SDW) anomalies, which may imply a different superconducting mechanism than previous FeAs-based superconductors. Its crystal structure is unique in that it has bilayer Fe${}_{2}$As${}_{2}$ layers [16]. At present, it has been found that a series of unique behaviors of 12442 may be related to these bilayer Fe${}_{2}$As${}_{2}$ layers, such as the T${}_{c}$ not being greatly affected by the purity [17] of the material; it not being greatly affected by the magnetic properties of lanthanide atoms; and the relationship of the T${}_{c}$ to the bond angle of Fe-As and the height of the As atom relative to the Fe layer is inconsistent with the empirical rules of the previous numerical study [18]. At present, Ghosh et al. have discussed the electronic properties of KCa${}_{2}$Fe${}_{4}$As${}_{4}$F${}_{2}$ [19], RbCa${}_{2}$Fe${}_{4}$As${}_{4}$F${}_{2}$, and CsCa${}_{2}$Fe${}_{4}$As${}_{4}$F${}_{2}$, and studies have shown that 12442 compounds have mixed multi-orbital and multi-band properties, and the contributions of these compounds are mainly from Fe-3D orbitals [20]. Unlike other hybrid iron-based superconductors, the contribution of As-4p${}_{z}$ orbitals is negligible in these compounds. The substitution of base atoms with progressively larger atomic radii exerts chemical pressure inside the compound and induces orbital-selective evolution of the band structure and density of states [20]. However, there is no report on the electronic properties of RbTb${}_{2}$Fe${}_{4}$As${}_{4}$O${}_{2}$ and RbDy${}_{2}$Fe${}_{4}$As${}_{4}$O${}_{2}$. At present, there are few articles on the electronic properties of RbGd${}_{2}$Fe${}_{4}$As${}_{4}$O${}_{2}$, and only the case of the nonmagnetic material without U is considered; the calculation results under magnetic conditions have not been considered [21].

In this paper, first principles methods, described in Section II, are applied to study the electronic structures and magnetic ground states of Rb*Ln*${}_{2}$Fe${}_{4}$As${}_{4}$O${}_{2}$. Section III contains the main results. It is confirmed that the ground states of Rb*Ln*${}_{2}$Fe${}_{4}$As${}_{4}$O${}_{2}$ are SDW-type, in-plane, striped antiferromagnets, and the magnetic moment around each Fe atom is about 2 $\mu_{B}$. Several other differences between the Gd, Tb, and Dy compounds were analyzed, including the effect of the chemical pressure brought about by their introduction on the FeAs conductive layer and the role of differential charge transfer in the structurally unique double FeAs layer of 12442. A brief summary is provided in Section 4.

## 2. Materials and Methods

### 2.1. Crystal Structure

For all the calculations, the following experimental lattice parameters [22] are used in the unit cell: *a* = 3.901 Å and *c* = 31.343 Å for RbGd${}_{2}$Fe${}_{4}$As${}_{4}$O${}_{2}$, *a* = 3.890 Å and *c* = 31.277 Å for RbTb${}_{2}$Fe${}_{4}$As${}_{4}$O${}_{2}$, and *a* = 3.879 Å and *c* = 31.265 Å for RbDy${}_{2}$Fe${}_{4}$As${}_{4}$O${}_{2}$ in nonmagnetic (NM) phase, ferromagnetic (FM) phases, and antiferromagnetic (AFM) phases. The shorter the Fe-As bond length, the stronger the metallicity, the stronger the ability to lose electrons, and the weaker the ability of the metal to control the outermost electrons. Therefore, the interaction between cations and free electrons becomes weaker, and the electron correlation decreases. The space group of NM, FM1, and FM2 are *I*4*/mmm* (No. 139), while the space group of AFM1, AFM2, AFM3 (AFM4), and AFM5 are *Cmma* (No. 67), *P*42*/mmc* (No. 131), *I*4*/mm* (No. 107), and *P*4*/mmm* (No. 123), respectively. Among them, AFM1 represents stripe antiferromagnetism (SAFM), that is, each Fe atom is aligned antiparallel to its second nearest neighbors along the a and b axes. AFM2 represents N$\overset{´}{e}$el antiferromagnetism (NAFM); that is, the spin directions of the nearest neighbor atoms in the Fe layer are opposite. AFM3, AFM4, and AFM5 are different interlayer antiferromagnets. The space group belongs to *P*4*/nmm*, where FeAs layers and *Ln*O layers are alternately arranged along the c-axis, forming a quasi-binary layered structure. The atoms in each layer are arranged in a four-coordinated tetrahedral structure and connected collinearly; in the FeAs layer, the 3D transition ferromagnetic atoms Fe form a two-dimensional square lattice, and charges are transferred from the *Ln*O layer to the FeAs layer. The FeAs layer balances the chemical bonds between the (*Ln*O)${}^{1+}$ layer and the (FeAs)${}^{1-}$ layer.

The structures of Rb*Ln*${}_{2}$Fe${}_{4}$As${}_{4}$O${}_{2}$ (*Ln* = Gd, Tb, and Dy) in the NM phase are shown in Figure 1. The As atoms in the FeAs bulk are no longer crystallographically equivalent. Here, the atomic coordinates are as follows: Rb (0, 0, 0); *Ln* (0.5, 0.5, z); Fe (0.5, 0, z); As1 (0.5, 0.5, z); As2 (0, 0, z); O (0.5, 0, 0.25). The antiferromagnetism of two *Ln* atoms is obtained by breaking the symmetry of the NM structure. As shown in Figure 1, we divide the FeAs layers in the electronic structure into an Fe${}_{2}$As${}_{2}$ intralayer and an Fe${}_{2}$As${}_{2}$ interlayer. It can be seen from the experimental results [22] that from Gd, Tb, to right Dy, both the Fe${}_{2}$As${}_{2}$ interlayer (Fe1As1) distance and the Fe${}_{2}$As${}_{2}$ intralayer (Fe2As2) distance become smaller, and these changes affect the superconducting transition temperature of the material.

### 2.2. Calculational Methods

The full-potential linearized augmented plane wave Wien2K package [23] was used for the electronic structure calculations. We used the Perdew, Burke, and Ernzerhof (PBE) [24] version of the generalized gradient approximation (GGA) within density functional theory. The sphere radii for Rb, Gd, Tb, Dy, and O were taken as 2.5 bohr, 2.33 bohr, 2.32 bohr, 2.32 bohr, and 1.9 bohr and as 2.3 and 2.19 bohr for Fe and As, respectively. The basis set cut-off parameter R${}_{mt}$·K${}_{max}$ = 7.0 was found to be sufficient, and the number of k points was 8000 for the tetragonal unit cell, 3000 for SAFM supercell, and 4500 for other supercells that were used for each of the calculations to provide an adequate sampling of small Fermi surfaces. The supercell shapes and spin distributions studied under all magnetisms are shown in Figure 2. Considering the strong correlation of the 4f electrons in Gd, Tb, and Dy, we applied the GGA+U (U = 9 eV) code for RbGd${}_{2}$Fe${}_{4}$As${}_{4}$O${}_{2}$, RbTb${}_{2}$Fe${}_{4}$As${}_{4}$O${}_{2}$, and RbDy${}_{2}$Fe${}_{4}$As${}_{4}$O${}_{2}$. We studied several references [25,26,27,28] and also calculated U = 6 eV and U = 12 eV at the same time. After comparison, it was found that U = 9 eV is more reasonable for these three materials.

## 3. Analysis and Discussion

We first present for the total energy difference of NM, FM, and several different SAFM configurations of RbTb${}_{2}$Fe${}_{4}$As${}_{4}$O${}_{2}$ and RbDy${}_{2}$Fe${}_{4}$As${}_{4}$O${}_{2}$, as well as for Fe in each magnetic phase. Table 1 shows the magnetic moments. We found that the SAFM configuration within the layer has the lowest total energy, so all of the ground states of Rb*Ln*${}_{2}$Fe${}_{4}$As${}_{4}$O${}_{2}$ are in the in-plane SAFM phase. The magnetic moment around each Fe atom is about 2 $\mu_{B}$, which is much smaller than the local density approximation (LDA) value in LaFeAsO [29]. As we will see later, DFT may be suitable for this compound due to the presence of bilayer Fe${}_{2}$As${}_{2}$ layers, Rb*Ln*${}_{2}$Fe${}_{4}$As${}_{4}$O${}_{2}$, which have better metallic properties and smaller magnetic moments than LaOFeAs. In the following, we will discuss the basic characteristics of the electronic structures of NM phases and the ground state in-plane SAFM in RbGd${}_{2}$Fe${}_{4}$As${}_{4}$O${}_{2}$, RbTb${}_{2}$Fe${}_{4}$As${}_{4}$O${}_{2}$, and RbDy${}_{2}$Fe${}_{4}$As${}_{4}$O${}_{2}$ and compare the differences between these three materials. We also analyze the effects of lanthanides and the role of bilayer Fe${}_{2}$As${}_{2}$ in detail.

### 3.1. Nonmagnetic Phase

Figure 3 shows the project of bands (pbands) of the NM phases Rb*Ln*${}_{2}$Fe${}_{4}$As${}_{4}$O${}_{2}$. We mainly discuss the energy bands contributed by Fe, and the band structure consists of four hole-like energy bands around the Γ(0, 0) point and four electron-like energy bands around the X(π,π) point. The NM band structures of Rb*Ln*${}_{2}$Fe${}_{4}$As${}_{4}$O${}_{2}$ are similar to those of 1111-type compounds such as LaFeAsO, which means that their ground states also tend toward an antiferromagnetic spin density wave state. The band dispersion of electrons along the Γ-Z direction of RbTb${}_{2}$Fe${}_{4}$As${}_{4}$O${}_{2}$ and RbDy${}_{2}$Fe${}_{4}$As${}_{4}$O${}_{2}$ is flat, showing considerable two-dimensional features, while the hole band structure of RbGd${}_{2}$Fe${}_{4}$As${}_{4}$O${}_{2}$ along the Γ-Z direction shows dispersion. It can be seen that only RbGd${}_{2}$Fe${}_{4}$As${}_{4}$O${}_{2}$ has an energy band contributed by Fe 3d${}_{z^{2}}$ near the Fermi level at the point closest to Γ. It shows that compared with Tb and Dy, Gd is more conducive to the occupation of electrons in the 3d${}_{z^{2}}$ orbital, which is beneficial to the interlayer charge transfer. This energy band also corresponds to the Fermi surface at the center of Figure 3d. Significantly different from the Fermi surfaces of conventional iron-based superconductors, such as the 1111 and 122 systems, there is an additional pocket of holes near the region center Γ(0, 0, 0). Compared with the three materials, only Gd has this Fermi surface. We further analyze the pDOS of Fe corresponding to five orbitals in the three materials, and we can find that in RbTb${}_{2}$Fe${}_{4}$As${}_{4}$O${}_{2}$, the three orbitals of 3d${}_{x^{2}-y^{2}}$, 3d${}_{xz}$, and 3d${}_{yz}$ are the main contributions at the Fermi energy and near the Fermi energy: 3d${}_{x^{2}-y^{2}}>$3d${}_{xz}>$3d${}_{yz}$. Similar to RbDy${}_{2}$Fe${}_{4}$As${}_{4}$O${}_{2}$, we can see that the main source of contribution near the Fermi level is also the three orbitals of Fe 3d${}_{x^{2}-y^{2}}$, 3d${}_{xz}$, and 3d${}_{yz}$, and the contributions of the three orbitals are in the same order, that is, 3d${}_{x^{2}-y^{2}}>$3d${}_{xz}>$3d${}_{yz}$. However, the difference in the DOS of each orbital is larger than that of RbTb${}_{2}$Fe${}_{4}$As${}_{4}$O${}_{2}$. From the pDOS map of the Fe of RbGd${}_{2}$Fe${}_{4}$As${}_{4}$O${}_{2}$, it can be found that it is obviously different from the orbital contribution of RbDy${}_{2}$Fe${}_{4}$As${}_{4}$O${}_{2}$ and RbTb${}_{2}$Fe${}_{4}$As${}_{4}$O${}_{2}$ near the Fermi level. It can be found in Figure 3a–c that for RbGd${}_{2}$Fe${}_{4}$As${}_{4}$O${}_{2}$, five orbits near the Fermi level contribute, and the main contribution comes from the 3d${}_{xz}$ of Fe, that is, 3d${}_{xz}>$3d${}_{yz}>$3d${}_{x^{2}-y^{2}}>$3d${}_{z^{2}}>$3d${}_{xy}$, in which the contribution of the 3d${}_{z^{2}}$ orbital is higher than the other two materials. It can thus be confirmed that the effect of Gd on 12442 is indeed different from that of Tb and Dy, and the presence of Gd is more conducive to the interlayer electron transfer.

#### 3.1.1. Minus *Ln*O Layers

From the Fermi surface structures of Rb*Ln*${}_{2}$Fe${}_{4}$As${}_{4}$O${}_{2}$ in the NM phase, it can be found that the Fermi surfaces of RbTb${}_{2}$Fe${}_{4}$As${}_{4}$O${}_{2}$ and the 1111 system are very similar, while the structures of RbDy${}_{2}$Fe${}_{4}$As${}_{4}$O${}_{2}$ are similar to the 122 system, so it is not difficult to discern the effect of the presence of *Ln* on these three materials. It is obviously different, so we calculated that if there is no rare earth element, we consider removing the Fermi surface of the *Ln*O layer and the difference map of charge density to obtain the different effects of Ln series elements Gd, Tb, and Dy in the 12442 iron-based superconductors. If the *Ln*O layer is removed, the electronic properties of Rb*Ln*${}_{2}$Fe${}_{4}$As${}_{4}$O${}_{2}$ should be very similar, and the only difference is the structural parameters.

Compared with the energy band structure before removing the *Ln*O layer, the obvious difference in Rb*Ln*${}_{2}$Fe${}_{4}$As${}_{4}$O${}_{2}$ is that the characteristic energy band at *X* point is moved up, as shown in Figure 4. However, Rb*Ln*${}_{2}$Fe${}_{4}$As${}_{4}$O${}_{2}$ looks very similar in general, indicating that the structural differences caused by the ionic radii have little effect on the properties of these three materials, and the different *Ln* elements themselves have a major impact on the properties of Rb*Ln*${}_{2}$Fe${}_{4}$As${}_{4}$O${}_{2}$.

#### 3.1.2. Dy vs. Gd and Tb

Considering that Fe-based compounds and chalcogenides occupy Fe orbitals and that their relationship with properties is the focus of attention, to further study the effects of Gd, Tb, and Dy atoms on Fe orbitals, we calculated the charge densities of Rb*Ln*${}_{2}$Fe${}_{4}$As${}_{4}$O${}_{2}$ in the NM phase. To facilitate the comparison, we have calculated the difference in density ρ(RbDy${}_{2}$Fe${}_{4}$As${}_{4}$O${}_{2}$)-ρ(RbGd${}_{2}$Fe${}_{4}$As${}_{4}$O${}_{2}$) and ρ(RbDy${}_{2}$Fe${}_{4}$As${}_{4}$O${}_{2}$)-ρ(RbTb${}_{2}$Fe${}_{4}$As${}_{4}$O${}_{2}$). To obtain the charge density of the two materials, both were calculated from the lattice constant of the previous compound. That is to say, the latter retains the lattice parameters and atomic positions of the former, but replaces the lanthanide elements to compare the effects of the two lanthanide elements on the electronic properties of the material.

Figure 5 shows a contour map of this differential density in two planes: the (110) plane containing the *Ln*O layer Fe${}_{2}$As${}_{2}$ intralayer (Fe2As2) and the Fe${}_{2}$As${}_{2}$ interlayer (Fe1As1). The unit of electron density is electrons per unit cell. Positive values indicate areas where Dy attracts charges relative to Gd and Tb. As can be seen from Figure 5, Gd increases the charge in the Fe 3d${}_{z^{2}}$-direction orbital relative to Dy, while the charge density in the z-direction of the *Ln*O layer decreases. This indicates that Gd contributes to the transfer of charge from the *Ln*O to the FeAs layer, confirming that previous Gd elements relative to Tb and Dy elements can play a role in enhancing the electron transfer between layers. While Tb reduces the charge in the Fe in-plane directional orbitals relative to Dy, the charge density in the z-direction of the *Ln*O layer decreases. In addition, we can find that the Fe${}_{2}$As${}_{2}$ intralayer and interlayer are different, especially as shown in ρ(RbDy${}_{2}$Fe${}_{4}$As${}_{4}$O${}_{2}$)-ρ(RbTb${}_{2}$Fe${}_{4}$As${}_{4}$O${}_{2}$). These differences can be calculated by subtracting the orbital occupancy within the atomic ball and comparing the orbital occupancy numbers of the individual atoms. RbDy${}_{2}$Fe${}_{4}$As${}_{4}$O${}_{2}$ minus RbTb${}_{2}$Fe${}_{4}$As${}_{4}$O${}_{2}$, within the atomic spheres. In units of 10${}^{-3}$, the differences for Fe are [d${}_{z^{2}}$,d${}_{x^{2}-y^{2}}$,d${}_{xy}$,d${}_{xz}$,d${}_{yz}$]:[−3.7,−5.1,−0.8,17.3,−16.3]. The net change for Fe is $-8.6\times 10^{-3}$ electrons. For As, the analogous changes are p${}_{x}$, −2.9; p${}_{y}$, −2.9; and p${}_{z}$, −1.2, for a small net change of $-6.9\times 10^{-3}$ electrons. RbDy${}_{2}$Fe${}_{4}$As${}_{4}$O${}_{2}$ minus RbGd${}_{2}$Fe${}_{4}$As${}_{4}$O${}_{2}$, within the atomic spheres. In units of 10${}^{-3}$, the differences for Fe are [d${}_{z^{2}}$,d${}_{x^{2}-y^{2}}$,d${}_{xy}$,d${}_{xz}$,d${}_{yz}$]:[−6.9,−4.2,−0.9,17.8,−15]. The net change for Fe is $-9.2\times 10^{-3}$ electrons. For As, the analogous changes are p${}_{x}$, −3.6; p${}_{y}$, −3.6; p${}_{z}$, −1.4, for a net change of $-8.5\times 10^{-3}$ electrons. Thus, Tb induces ∼0.016 electrons into the Fe + As spheres compared to Dy, while Gd induces ∼0.018 electrons into the Fe + As spheres compared to Tb, with the decrease occurring in the interstitial region or the (Dy, Tb)O or (Dy, Gd)O regions, respectively. This shows that Gd can transfer more electrons from the GdO layer to the FeAs layer than Tb and Dy.

### 3.2. Ground State: In-Plane SAFM Phase

Here, we discuss magnetic energies and the basic features of the electronic structure of Rb*Ln*${}_{2}$Fe${}_{4}$As${}_{4}$O${}_{2}$ and draw some parallels with related pnictides. The energies of seven of the simplest magnetic configurations relative to the FM1 phase are presented in Table 1, where the Fe atomic sphere moments are also provided. Because the energy of the NM state is much higher than that of the magnetic state, the difference is too large to be conducive to comparison. Therefore, the FM1 ferromagnetic state is used as the reference state. The DFT methods [30] overestimate the Fe moment, and thus the magnetic energy will also be exaggerated, but the relative ordering is still meaningful.

It can be obtained from the table that the ground states of Rb*Ln*${}_{2}$Fe${}_{4}$As${}_{4}$O${}_{2}$ are all in-plane SAFM (AFM1), and their corresponding structures are shown in the Figure 2. Among them, the magnetic moment of Fe is almost 2 $\mu_{B}$, which is very close to the 1.92 $\mu_{B}$ magnetic moment of the CaFeAsH [31] ground-state antiferromagnetism calculated by Wien2K before, and the magnetic moment of 2.11 $\mu_{B}$ of the CaFeAs${}_{2}$ [32] ground-state antiferromagnetic force is also very close. The magnetic moments of Fe determined by the neutron scattering experiment for the underdoped samples of LaFeAsO and BaFe${}_{2}$As${}_{2}$ are 2.38 $\mu_{B}$ [29] and 1.6 $\mu_{B}$ [33], which are greater than the ordered magnetic moment, indicating that the magnetic fluctuation is strong even in the matrix.

Figure 2 shows a magnetic structure diagram; from left to right are FM1, FM2, AFM1, AFM2, AFM3, AFM4, and AFM5 phases. Here, the coordinates of Fe1 and Fe2 are (0, 0, z) and (0.5, 0.5, z), and Ln1 (0.5, 0.5, z) and Ln2 (0, 0, z) are the Ln of different layers. As mentioned above, AFM1 is SAFM, AFM2 is NAFM, and AFM3, AFM4, and AFM5 are different interlayer antiferromagnets. The corresponding AFM1 energy in Table 1 is the lowest, which means that the magnetic ground states of these three materials are in the SAFM state.

Figure 6 shows the Fe pDOS of Rb*Ln*${}_{2}$Fe${}_{4}$As${}_{4}$O${}_{2}$ in the SAFM state. These three materials exhibit distinct electronic structural features in the SAFM state. The five spin-up orbitals are almost fully occupied, with complete electron occupation, while the spin-down orbitals are only partially filled, with an equal distribution of electrons and holes. The Fermi level contribution is mainly concentrated in the spin-down orbitals, where the Fe-3d${}_{xz}$ orbital has the largest contribution to the Fermi level, followed by Fe-3d${}_{z^{2}}$. Compared to Rb*Ln*${}_{2}$Fe${}_{4}$As${}_{4}$O${}_{2}$ (*Ln* = Gd and Tb), the contributions of the Fe-3d${}_{xz}$ and Fe-3d${}_{z^{2}}$ orbitals in RbDy${}_{2}$Fe${}_{4}$As${}_{4}$O${}_{2}$ are reduced by approximately 0.1 eV.

Figure 7 shows the Fe atoms pbands of RbGd${}_{2}$Fe${}_{4}$As${}_{4}$O${}_{2}$, RbTb${}_{2}$Fe${}_{4}$As${}_{4}$O${}_{2}$ and RbDy${}_{2}$Fe${}_{4}$As${}_{4}$O${}_{2}$, respectively. The corresponding Fermi level changes in the three materials near the *T* and *Z* positions are more obvious. The projected band structure shows that two energy bands, Fe-3d${}_{x^{2}-y^{2}}$ and Fe-3d${}_{yz}$ and Fe-3d${}_{z^{2}}$ contribute to the band crossing the Fermi level. From the left to the right of the figure, it can be clearly found that the energy bands near the Fermi level move up sequentially, among which the energy bands of RbGd${}_{2}$Fe${}_{4}$As${}_{4}$O${}_{2}$ are mainly contributed by Fe-3d${}_{yz}$ and Fe-3d${}_{z^{2}}$ orbitals, while the energy bands of RbDy${}_{2}$Fe${}_{4}$As${}_{4}$O${}_{2}$ are mainly composed of Fe-3d${}_{x^{2}-y^{2}}$. It can be shown that Gd is more conducive to the electron transfer between layers than Tb and Dy, which is consistent with the conclusion drawn under the nonmagnetic phase.

## 4. Discussion and Summary

For the first time, we have studied the high-temperature nonmagnetic state and the low-temperature magnetic ground state of three 12442 materials, RbTb${}_{2}$Fe${}_{4}$As${}_{4}$O${}_{2}$, RbDy${}_{2}$Fe${}_{4}$As${}_{4}$O${}_{2}$, compared them with RbGd${}_{2}$Fe${}_{4}$As${}_{4}$O${}_{2}$, and performed a detailed study of the effects of lanthanides and bilayer Fe${}_{2}$As${}_{2}$. It was found that the two 12442 materials RbTb${}_{2}$Fe${}_{4}$As${}_{4}$O${}_{2}$ and RbDy${}_{2}$Fe${}_{4}$As${}_{4}$O${}_{2}$ have the same antiferromagnetic ground state. The ground state of Rb*Ln*${}_{2}$Fe${}_{4}$As${}_{4}$O${}_{2}$ are spin-density-wave-type, in-plane, striped antiferromagnets, and the magnetic moment around each Fe atom is about 2 $\mu_{B}$. We also found that the structural differences caused by the simple ionic radius have little effect on the properties of these three materials. The electronic properties of the materials are strongly influenced by the specific lanthanide element present. It can be confirmed that the effect of Gd on Rb*Ln*${}_{2}$Fe${}_{4}$As${}_{4}$O${}_{2}$ is indeed different from that of Tb and Dy, and the presence of Gd is more conducive to interlayer electron transfer. This means that Gd can transfer more electrons from GdO layer to FeAs layer compared to Tb and Dy. Therefore, RbGd${}_{2}$Fe${}_{4}$As${}_{4}$O${}_{2}$ has a stronger internal coupling strength of the bilayer Fe${}_{2}$As${}_{2}$ layer. This can explain why the T${}_{c}$ of RbGd${}_{2}$Fe${}_{4}$As${}_{4}$O${}_{2}$ is slightly higher than that of RbTb${}_{2}$Fe${}_{4}$As${}_{4}$O${}_{2}$ and RbDy${}_{2}$Fe${}_{4}$As${}_{4}$O${}_{2}$. We hope that these findings will help to study such materials and understand their relationship with superconducting properties, thus providing theoretical support for searching for superconducting materials with higher T${}_{c}$.

## Figures and Tables

**Figure 1 materials-16-04123-f001:**
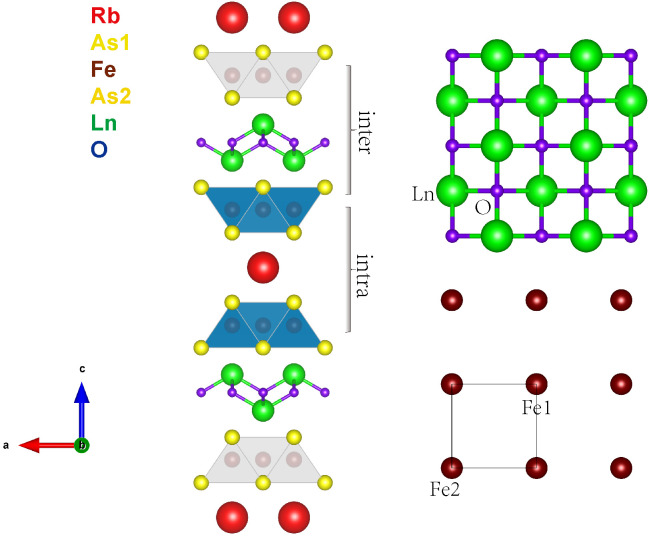
(Color online) The structures of Rb*Ln*${}_{2}$Fe${}_{4}$As${}_{4}$O${}_{2}$ (*Ln* = Gd, Tb, and Dy) in NM phase, including the structural diagrams of the lanthanide layer and the iron atomic layer.

**Figure 2 materials-16-04123-f002:**
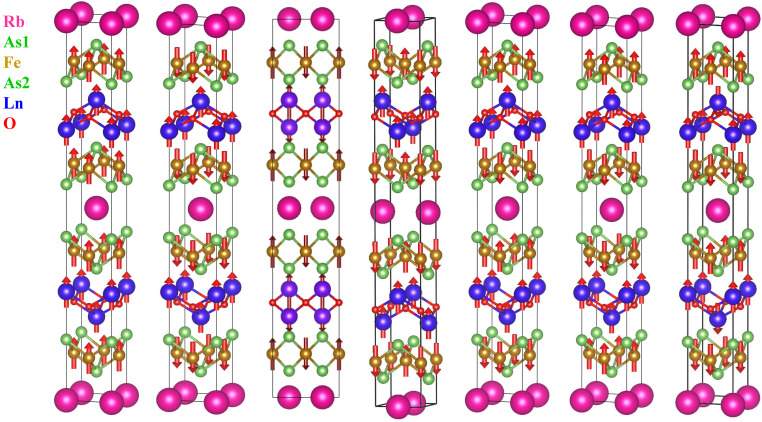
(Color online) The structures of Rb*Ln*${}_{2}$Fe${}_{4}$As${}_{4}$O${}_{2}$ (*Ln* = Gd, Tb, and Dy) in FM1, FM2, AFM1, AFM2, AFM3, AFM4, and AFM5 phases.

**Figure 3 materials-16-04123-f003:**
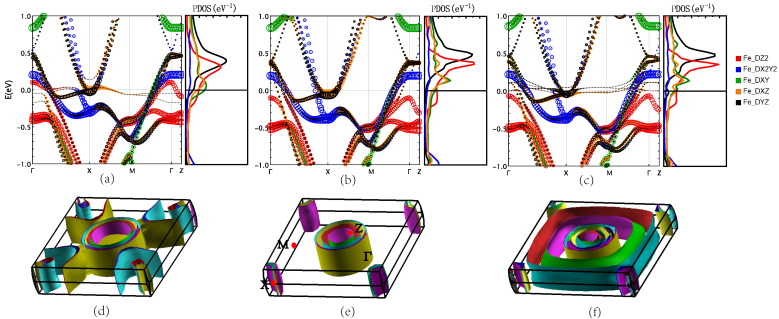
(Color online) The Fe projection of bands, projected density of states, and the Fermi surfaces of (**a**,**d**) RbGd${}_{2}$Fe${}_{4}$As${}_{4}$O${}_{2}$, (**b**,**e**) RbTb${}_{2}$Fe${}_{4}$As${}_{4}$O${}_{2}$, and (**c**,**f**) RbDy${}_{2}$Fe${}_{4}$As${}_{4}$O${}_{2}$ in NM phase.

**Figure 4 materials-16-04123-f004:**
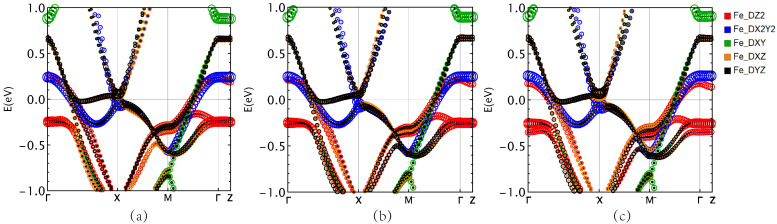
Electronic band structure along high symmetry line Γ−X−M−Γ−Z of (**a**) RbGd${}_{2}$Fe${}_{4}$As${}_{4}$O${}_{2}$ minus GdO layer, (**b**) RbTb${}_{2}$Fe${}_{4}$As${}_{4}$O${}_{2}$ minus TbO layer, and (**c**) RbDy${}_{2}$Fe${}_{4}$As${}_{4}$O${}_{2}$ minus DyO layer in NM phase.

**Figure 5 materials-16-04123-f005:**
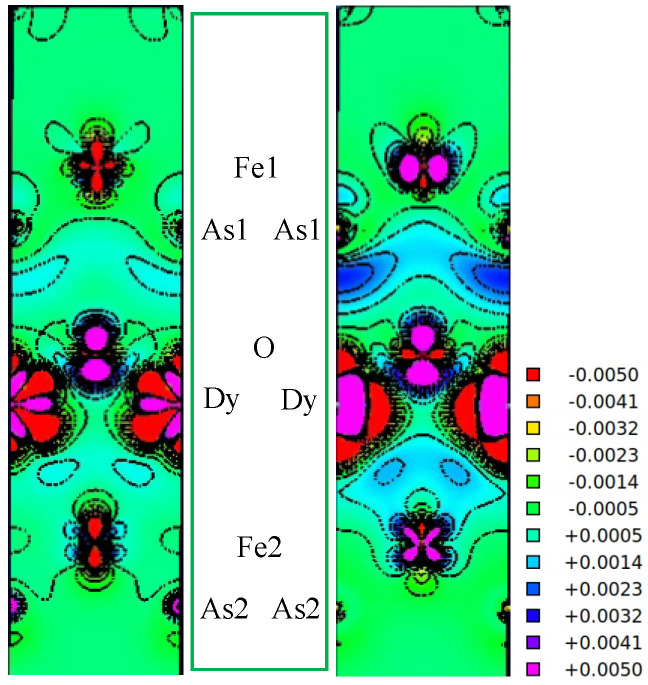
Contour plots of the density difference ρ(RbDy${}_{2}$Fe${}_{4}$As${}_{4}$O${}_{2}$)-ρ(RbGd${}_{2}$Fe${}_{4}$As${}_{4}$O${}_{2}$) and ρ(RbDy${}_{2}$Fe${}_{4}$As${}_{4}$O${}_{2}$)-ρ(RbTb${}_{2}$Fe${}_{4}$As${}_{4}$O${}_{2}$). The large red region denotes where contours have been cut off due to the large and meaningless difference of the RbDy${}_{2}$Fe${}_{4}$As${}_{4}$O${}_{2}$ and Rb(Gd/Tb)${}_{2}$Fe${}_{4}$As${}_{4}$O${}_{2}$ densities in NM phase.

**Figure 6 materials-16-04123-f006:**
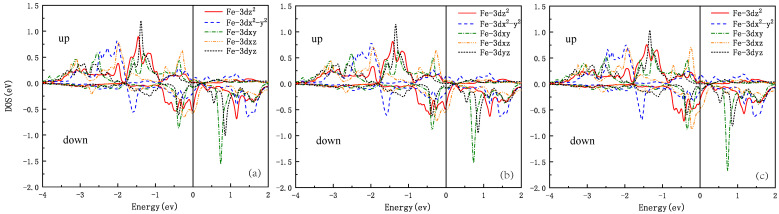
For Rb*Ln*${}_{2}$Fe${}_{4}$As${}_{4}$O${}_{2}$ with SAFM order: (**a**) RbGd${}_{2}$Fe${}_{4}$As${}_{4}$O${}_{2}$, (**b**) RbTb${}_{2}$Fe${}_{4}$As${}_{4}$O${}_{2}$, and (**c**) RbDy${}_{2}$Fe${}_{4}$As${}_{4}$O${}_{2}$ projected DOS for the Fe atom. Note the small DOS at the Fermi energy (zero of energy).

**Figure 7 materials-16-04123-f007:**
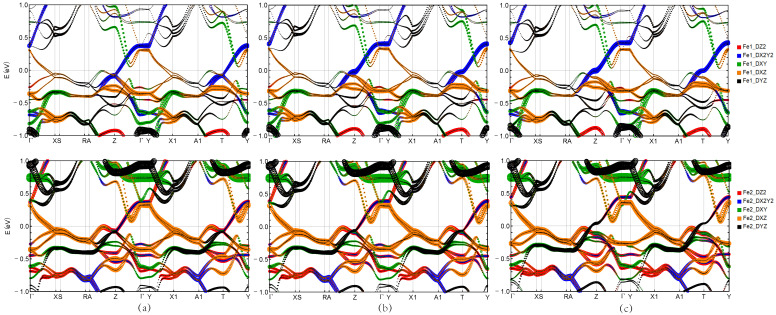
For Rb*Ln*${}_{2}$Fe${}_{4}$As${}_{4}$O${}_{2}$ with SAFM order: (**a**) RbGd${}_{2}$Fe${}_{4}$As${}_{4}$O${}_{2}$, (**b**) RbTb${}_{2}$Fe${}_{4}$As${}_{4}$O${}_{2}$, and (**c**) RbDy${}_{2}$Fe${}_{4}$As${}_{4}$O${}_{2}$ projected band structure for the Fe atom. Upper row: spin up. Lower row: spin down.

**Table 1 materials-16-04123-t001:** Total energy difference, for Rb*Ln*${}_{2}$Fe${}_{4}$As${}_{4}$O${}_{2}$ (*Ln* = Gd, Tb, and Dy) of seven magnetic phases including FM1, FM2, AFM1, AFM2, AFM3, AFM4, and AFM5 phases. The reference is the energy of the FM1 phase (ΔE = E${}_{(S)FM}-E$${}_{FM1}$). The corresponding magnetic moment in the Fe sphere is given.

	Magnetic Structure	FM1	FM2	AFM1	AFM2	AFM3	AFM4	AFM5
(a) Relative energy (meV)	RbGd${}_{2}$Fe${}_{4}$As${}_{4}$O${}_{2}$	0	−11	−429	−239	−6	−27	−8
	RbTb${}_{2}$Fe${}_{4}$As${}_{4}$O${}_{2}$	0	−81	−478	−218	156	−44	−6
	RbDy${}_{2}$Fe${}_{4}$As${}_{4}$O${}_{2}$	0	−872	−1236	−1083	−879	−874	−877
(b) Fe moment ($\mu_{B}$)	RbGd${}_{2}$Fe${}_{4}$As${}_{4}$O${}_{2}$	0.80	0.83	2.02	2.07	0.85	0.81	0.84
	RbTb${}_{2}$Fe${}_{4}$As${}_{4}$O${}_{2}$	0.81	0.83	1.98	2.06	0.90	0.84	0.87
	RbDy${}_{2}$Fe${}_{4}$As${}_{4}$O${}_{2}$	0.91	0.84	1.91	2.03	0.86	0.84	0.90

## Data Availability

The data presented in this study are available on request from the corresponding author.

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
