# Peer review of "Role of Lanthanides and Bilayer Fe2As2 in the Electronic Properties of RbLn2Fe4As4O2 (Ln = Gd, Tb, and Dy) Superconductors"

_materials, 2023, doi:10.3390/ma16114123_

Round 1
Reviewer 1 Report
The manuscript was originally submitted to Nanomaterials but, upon revision, has now been transferred to Materials. First of all, this seems a more appropriate choice, as RbLn2Fe4As4O2, although layered, are periodic bulk materials whose properties do not depend on nanosize particle dimensions. Many of the recommendations from the first review were also incorporated in the resubmitted version. In particular, the list of references has been expanded and now gives a more comprehensive overview of relevant publications, explanations have been rephrased and thereby improved in many places, and the objectives and conclusions are better clarified. In the process, the length of the manuscript was also reduced from 14 to 11 pages, chiefly by drastically shortening the discussion of differences in the charge density in Section 3.2 as Ln varies from Gd to Tb and Dy, together with the removal of the previous Figures 9 and 10. As a consequence, the manuscript now reports less calculated data, but this is in fact an improvement for readers, as the previous discussion was somewhat tedious to follow and did not actually strengthen the arguments but rather served as a diversion. In the present version, the same point is fully made in Section 3.1.2 for the nonmagnetic state, which allows a more concise discussion, and need not be repeated at the same level of detail for the magnetic configuration.
Overall, the quality of presentation has sufficiently improved so that I can now recommend to proceed with publication. The manuscript reports novel results with adequate conclusions that should be interest for researchers concerned with the design and fabrication of iron-based superconductors, and it deserves a place in the scientific literature.
As part of the further editorial process, moderate English editing will still be necessary. Furthermore, I recommend the authors to consider the following minor points before releasing the final proofs:
- The transition temperature of RbDy2Fe4As4O2 is given as 34.3 K in line 2 but as 33.8 K in line 57. Please check which value is correct and make changes where necessary.
- In the paragraph between lines 47 and 59, KCa2Fe4As4F2 is cited as an example of Ln3Fe4X4Z2, although neither K nor Ca are lanthanides (Ln). Likewise, AkAeFe4As4 and AkEuFe4As4 are cited as examples of KLaFe4X4, although they contain elements other than K and La, even alkaline earth metals (Ae) instead of lanthanides. The authors should check that the designations of structures and experimentally realized materials in this section are consistent. In the same context, A in line 60 should be changed to Ak for consistency.
- The characterization of AFM1 (SAFM) and AFM2 (NAFM) in lines 100 to 104 is identical, both are described in exactly the same words as "the spin direction of the nearest neighbor atoms in the Fe layer is opposite". The authors should rephrase the text in order to make the distinction clear to readers.
- In the revised Figure 1, Rd must be changed to Rb.
- Most of the figures are set too small in the reviewers' version of the manuscript. While some, such as Figures 1 or 4, can be easily enlarged to the width of the text column, this is not possible for Figures 2, 3, 6, and 7. For these four figures, it may be better to arrange the panels for the three materials not horizontally but vertically, so that they become longer in vertical length and shorter in horizontal width, which allows a subsequent enlargement. In the case of Figure 2, the Fermi surfaces would then be placed to the right of the band structures rather than below, as would the two spin channels in Figure 7. In Figure 2, the horizontal axes of the pDOS should be labeled "pDOS (eV^-1)", so that the meaning of the curves is evident from the figure itself and not only from the caption or main text.
- In lines 219 to 225, it appears that all five d orbitals are counted for the net change for Fe, but only the py and pz orbitals are counted for As while px is ignored. The authors should check the numbers and correct them if necessary.
- According to line 235, "the energy of the NM state is much lower than that of the magnetic state", but this would mean that the NM and not the magnetic state is the energetically favored ground state. Presumably the authors mean "much higher" instead of "much lower"?
The level of English is appropriate for the review stage but should still be properly edited before publication. In general, there are only few misspellings, possibly due to the informed use of automatic spellchecking, but occasional grammatical errors, such as incomplete sentences (example: "In contrast to the band structure without traditional iron-based materials." with no verb in lines 155-156) or mismatched singular and plural forms (example: "this bilayer Fe2As2 layers" in lines 65-66), as well as frequent incorrect choices of words (examples: "reduced state" instead of "reference state" in line 236, "are mainly contributed by" instead of "are mainly composed of" in line 270).
Reviewer 2 Report
All my comments have been thoroughly addressed
Author Response
Thank you very much.
Reviewer 3 Report
The authors study the 12442 materials in the low and high temperature regime by focusing on the role of lanthanides and of the bilayer block of the type Fe_2As_2. By means of first principle calculations they have investigated the ground state and found to be a stripe antiferromagnet. They also consider the role of the lanthanides in modifying the electronic structure and the hybridization between LnO and FeAs layers. The results are clearly presented, the methodology is well described, and the outcomes are interesting for the 12442 family of iron based superconductors.
I would like to suggest to the authors to check the use of singular and plural in the sentences.
For instance, in the abstract "the ground state ...are" should be "the ground state is".
Line 81: "first principles methods ... is" should be "first principles methods ...are" .
Please check through the whole manuscript.
Author Response
Dear reviewer,
Thank you very much for your suggestion.
We have checked the entire article according to your suggestion, corrected the two places you raised, and modified other places where singular and plural errors were used.
Sincerely,
Yina Huang
Reviewer 4 Report
It was shoen that the high-temperature non-magnetic state and the low-temperature magnetic ground state of three 12442 materials, RbTb2Fe4As4O2, 275 RbDy2Fe4As4O2, and comparing with RbGd2Fe4As4O2, and made a detailed study of the effects of lanthanides and bilayer Fe2As2. Also established that the two 12442 materials RbTb2Fe4As4O2 and RbDy2Fe4As4O2 have the same antiferromagnetic ground state. Ground state of RbLn2Fe4As4O2 are spin-density-wave-type in-plane striped antiferromagnets, and the magnetic moment around each Fe atom is about 2 µB.
It was shown that, the lanthanide elements themselves play a major role in the electronic properties of the compounds. It can be confirmed that the effect of Gd on RbLn2Fe4As4O2 is indeed different from that of Tb and Dy, and the presence of Gd is more conducive to interlayer electron transfer.
In my opinion, results obtained in this study will help to understand such compunds and their relationship with superconducting properties. This moment will be useful for the theoretical support for searching for superconducting materials with higher Tc.
In my opinion, presented paper can be accepted for publication. Before publishing it will be good to correct any misspoints in paper. Also it is useful to add to list of references book related with new class of superconductors:
Askerzade I., Unconcentional Superconductors: anisotropy and multiband effects, 2012, (Springer, Berlin), 177 p.
Author Response
Dear reviewer,
Thank you very much for your suggestion.
According to your suggestion, we have checked the article and corrected the errors. Thank you very much for your suggested citations for reference books, which we have cited in the text.
Sincerely,
Yina Huang